# Polypropylene Recovery and Recycling from Mussel Nets

**DOI:** 10.3390/polym14173469

**Published:** 2022-08-25

**Authors:** Loris Pietrelli

**Affiliations:** Department of Chemistry, Sapienza University, P.le A. Moro, 00185 Rome, Italy; loris.pietrelli@uniroma1.it

**Keywords:** mussel nets, circular economy, mussel farms, polypropylene, life MUSCLES

## Abstract

Mussels represent about one-third of all aquaculture products sold in the European Union. Theoretically, mussel production should be an environmentally friendly and sustainable activity (0.252 kg CO_2_ eq. per 1 kg of mussel produced against over 20 kg CO_2_ eq. per 1 kg of beef produced) but the abandoned plastic “socks” on the seabed and along beaches represent a significant environmental problem. The recovery and recycling of those polymer materials represents the proper management of the waste issue due to mussel farming. This study was performed to investigate, for the first time, the roles of the chemical oxidation actions on the detachment (and destruction) of organic matter (biofilm in particular) from the surface of the polypropylene “socks” used in sea farms in order to recover the polymer material and recycle it. In the experiments, oxidation by H_2_O_2_ and HNO_3_ was performed on the studied samples. The effects of the particle size of the fragments, oxidant concentration, agitation time and ultrasound application were determined. FTIR spectra and tensile mechanical properties of the samples after treatment were measured and compared with the virgin polymer material. The biodiversity and structure of the plastic-associated biofilm was also determined before and after the oxidation process. Based on the results of the characterization of the recovered polymer material, a process scheme was designed. The application of the developed process could significantly reduce the environmental risk associated with used mussel socks. The One LIFE (the EU’s funding instrument for the environment and climate action) Project was recently founded based on this research.

## 1. Introduction

Aquatic products, mainly fish, molluscs and crustaceans, have a critical role in the food system, providing nearly 3 billion people with at least 15% of their animal protein consumption [1]. Mussels represent about one-third of all aquaculture products sold in the European Union, reaching 522,400 tonnes in 2016, 24.5% in respect to the world production. Regarding the cultivated species, in the Atlantic Ocean, the blue mussel *Mytilus edulis*, is the main focus of production, while in the Mediterranean and the Black Seas, the Mediterranean mussel *Mytilus galloprovincialis* is the major cultivated species [2]. 

In sea farms, molluscs grow in the same way as animals that live spontaneously on marine rocks. Therefore, it is not necessary to feed them specifically because they are fed by filtering the plankton from seawater, contributing to the removal of nutrients [3]. It has been estimated that in 2015, cultivated bivalves removed 49,000 tonnes of nitrogen and 6000 tonnes of phosphorus [4].

Theoretically, mussel production should be an environmentally friendly and sustainable activity; in fact, the carbon footprint of mussel production was calculated to be 0.137−0.252 kg CO_2_ eq. per 1 kg of produced mussels against 8.7 kg CO_2_ eq. per 1 kg of produced oysters or over 20 kg CO_2_ eq. per kg of produced beef [5,6,7]. No land-use, no greenhouse gas emissions and no freshwater are used or polluted as a result of mussel farming. Despite these positive facts, there are two main problems concerning this production activity: (1) the deposition of a large amount of suspended matter, such as faeces and pseudo-faeces, which can have impact on surrounding communities [8]; and (2) a completely different, but no less important, problem concerns marine litter, in particular the “socks” used in mussel farming and abandoned on the seabed and along the beaches. The Mediterranean Sea has been considered as one of the areas most affected by marine litter in the world (UNEP/MAP 2015) [9], contributing strongly to beach litter [10]. According to Strafella et al. [11], lost fishing nets and debris from the cultivation of mussels account for 50% of the overall plastic litter collected, especially in the areas close to sea farms along the Adriatic coast. These plastic items that have an estimated lifetime of hundreds of years before their complete mineralization can be dispersed far from the pollution sources, causing damage to marine biodiversity and ecosystems. In an investigation regarding the seabed activity, the presence of nets was the highest (73 socks per square km of seabed) [12]. 

Culture activity begins with the collection of wild seed mussel either from natural beds or from a rope or another collector placed in areas chosen according to their characteristics such as movements of the seawater (currents) and the presence of microorganisms [13]. The ropes are collected and transferred to mussel farms, generally between May and July. After approximately 9−12 months of cultivation, the mussels triple their weight and reach a size suitable for marketing (6−10 cm). Usually, the mussel harvest takes place throughout the year, although there is a period of greater intensity between October and March, when about 70% of the mussels arrive on the market.

In the Mediterranean Sea, the most common rearing method developed by mussel farmers is the long-line system that produces high quality mussels per unit surface [14]. The long-line culture system is composed of a 50−60 m main line made of polypropylene rope that remains anchored to the seabed by concrete blocks (Figure 1). Attached to the main line are socks containing the growing bivalves molluscs. Mussel socks (3−5 m long) are tied to the long-line at regular intervals (50 cm) and cultivation takes place between 3 and 7 m deep [14].

During the life cycle of the mussel, the nets are replaced twice and, especially during the change, since the operation is carried out “on site”, part of the socks are dispersed at sea due to accidents, storm surges, distraction or, simply, bad behaviour by the operators themselves. 

Basically, for each kg of mussels produced, about one linear meter of polypropylene (PP) net is used. The 2016 production of the *M. galloprovincialis* in the Mediterranean Sea reached 318,905 tonnes [15]; this means a hypothetical 6378 tonnes of plastic waste could be considered produced in one year (20 g of PP/m of net). Furthermore, according to some local producers, this value could be underestimated due to the higher unofficial production.

In Italy, every year, over 70,000 tons of mussels are sold; this amount to about 1500 tons of nets, mostly in polypropylene, used in marine farms to support mussels during their growth until they reach the size necessary to sell the mollusc. 

The EU Commission’s proposals on Fisheries control COM (2018) 368 final 2018/0193 (COD) [16] and the EU Strategy for Plastics in a Circular Economy [17] are the first EU-wide policy frameworks to adopt both a fishing sustainability and a material-specific lifecycle approach to integrate circular design, use, reuse and recycling activities into plastics value chains. Moreover, the Commission has introduced circularity aspects in terms of energy consumption and material use, waste prevention, recycling and reduction in hazardous chemicals in specific Best Available Techniques Reference Documents (BREFs). These important inputs should improve the management of coastal areas from a sustainable development viewpoint, even if it creates conflict due to the unavoidable overlapping of economic and ecological interests.

Sound and efficient waste management systems are an essential building block of a circular economy; therefore, it is necessary to initiate the proper management of mussel nets through effective collection, which must be followed by a treatment system oriented towards the recovery and recycling of the polymer material used for mussel nets production.

Due to the presence on the surface of incrustation of salty biological elements and fouling, currently, a European Waste Code (CER 020104) of special waste is associated with the mussel socks: their correct disposal involves significant expenses (0.20–0.25 EUR/kg). Surprisingly, the nets have always been considered “non-recyclable” and to our knowledge, there is no bibliographical reference regarding the topic, so it was not possible to make any comparison, regarding the management, with the literature data.

This study was performed to evaluate, for the first time, the effectiveness of different methods used to remove adhered particles and improve the cleanliness level of the plastic surface in order to recover and recycle polymer material, polypropylene in particular.

## 2. Materials and Methods

All chemicals, hydrogen peroxide (H_2_O_2_), nitric acid (HNO_3_) and sodium hydroxide (NaOH), of reagent grade were supplied by Merck (Merck KGaA, Darmstadt, Germany) and were used without any further purification. Deionized water was used to prepare all solutions utilised during the experimental tests.

Net samples (new and used) were taken by AMA (Mediterranean Aquaculture Association, the largest Italian mussel farming enterprises association) directly from the mussel farms located in the Adriatic Sea; small samples of nets used to compare the quantity of adhered solid material only were from the Ligurian Sea.

The samples were kept in the refrigerator for as long as necessary to conduct the experimental tests. To evaluate the total quantity of organic and inorganic materials adhered on the plastic surface, twenty samples (5 g each), as such, were placed in the oven (ArgoLab TCN 30, G. Bormac Srl, Modena, Italy) ) at 60 °C until a constant weight was reached and subsequently treated repeatedly with a strong oxidant (H_2_O_2_ at 40% in the presence of Fe^2+^ ions) washed, dried and weighed to determine the weight loss produced by the chemical treatment. The dry matter adhered on the plastic surface was calculated as follows:Removal % = [(Wi − Wf)/Wc] ∗ 100(1)
where Wi = initial weight, Wf = final weight and Wc = clean weight.

Considering the high inhomogeneity in the distribution of the adhered material on the plastic surface, it was not possible to establish a standard contamination value to use as a reference during the experimental tests. Hence, the quantity of organic and inorganic materials adhered on the plastic surface was determined for each sample utilized in the experiment. Furthermore, a large sample (50 g) was divided into 10 subsamples, where three samples were used to estimate the total adhered material and the remaining samples were used for the cleaning tests. To properly evaluate the effect of the cleaner solution the role of the oxidant agent (H_2_O_2_ and HNO_3_) concentration, treatment time, pH, Fe^2+^ concentration, sample dimensions and ultrasound application on the removal efficiency were determined. The oxidant solutions were prepared using distilled water, the samples (about 5 g each) taken from the oven-dried nets were immersed in 200 mL of solution, the solutions were stirred for 30 min at 200 rpm. Each experimental test was repeated three times. To evaluate the adhered dry materials removed by mechanical action during the crushing step, 100 g of dry socks were crushed to obtain different sizes (<5 mm, 1 × 1 cm and 5 × 5 cm) and the obtained samples were properly sieved to separate dry matter from plastics. The dry matter removed by crushing was calculated as the percent of dry socks.

Observations with an optical stereomicroscope (LEICA KL200 LED, Leica Microsystems GmbH, Wetzlar, Germany) were performed in order to check the efficiency removal of the adhered materials on the plastic surface.

The rate of material removal from the plastic surface was evaluated by applying the first order kinetic model:ln (W_t_/W_0_) = −kt (2)
where W_t_ represents the net weight at time t, while W_0_ is the initial weight and k the velocity constant.

Spectrophotometer determination of H_2_O_2_ concentration was performed using toluidine blue method at 628 nm [18]. Spectrophotometric measurements were carried out using a Shimadzu 1900i UV−VIS instrument.

The polymer characterization was performed using a non-destructive Attenuated Total Reflection (ATR) mode Fourier Transform Infrared spectroscopy (FT-IR) using Nicolette 6700 spectrophotometer (Thermo Fisher Scientific, Waltham, MA, USA); all spectra were obtained in the absorbance mode in the 4000−600 cm^−1^ region with 4 cm^−1^ resolution and 50 scans. The spectroscopic bands utilized for Polypropylene identification were the following: 2800–2950 cm^−1^, 1450−1480 cm^−1^, 1350−1370 cm^−1^, 1020−1080 cm^−1^, 800−850 cm^−1^, and 690−740 cm^−1^. To analyse any oxidegradation process, three regions of the middle infrared spectrum were considered: the range of the O-H stretching modes (3200−3600 cm^−1^), the range of the C=O stretching modes (1600–1850 cm^−1^), the aliphatic C-O-C absorption at 1150–1085 cm^−1^ region due to asymmetrical stretching and the range of the rocking of the C=C bonds (850–1050 cm^−1^). 

The integrity of the recycled PP was determined by TG measurements carried out by a Mettler Toledo thermogravimetric analyser (Mettler Toledo, Columbus, OH, USA) under a nitrogen atmosphere (gas flow = 20 mL min^−1^). The heating rate was fixed at 5.0 °C min^−1^ and the samples (5–6 mg) were placed in an alumina crucible.

Mechanical properties of the recycled PP were compared with virgin polymer material by the tensile strength of the nets determined using a uniaxial tensile testing device (INSTRON 4502, Instron Inc. Norwood, MA, USA) equipped with a 10 N load cell and the crosshead speed was 1 mms^−1^. Each result was taken from three rectangular samples (60 mm × 10 mm) according to the ASTM D882-02 standard methodology for Tensile Properties of Thin Plastic Sheeting.

The morphology of the adhered material was assessed by Scanning Electron Microscopy (SEM) using a LEO 1530 (Zeiss, Oberkochen, Germany) instrument. The images were obtained using an accelerating voltage of 10−15 kV and samples were sprayed with a fine layer of gold using a low deposition rate. 

According to the experimental procedure previously adopted [19], 16S rRNA gene high throughput sequencing and Fluorescence in Situ Hybridization (FISH) combined with confocal laser scanning microscopy (CLSM) were used to assess the biodiversity and structure of biofilms (plastisphere) adhered on the surface of the socks. DNA extraction was performed on biofilms using the PowerSoil^®^DNA isolation Kit (MoBio-Carlsbad, CA, USA). To check the efficiency of the oxidation process, the microbial community on the socks was characterized before and after the treatment. 

## 3. Results and Discussion

### 3.1. Socks Characterization

Characterization of hazardous waste is key to improve the decontamination techniques, which facilitate the recovery and recycling of materials. In particular, mussel nets are characterized by the presence of calcium carbonate from calcifying marine invertebrates, byssus and microorganisms (biofilm) that are closely attached to the polymeric surface and biological matter from died mussels, as clearly illustrated in Figure 2. 

The weight loss of the socks derived from their simple drying at 60 °C in the oven varies in the range 11.6−20.8% in relation to the quantity and type of adhered materials and of the water present, especially in the biofilm. Obviously, these are only purely indicative values since the samples were taken several days before the treatment tests and therefore could have lost part of the water and material adhered during the transport or storage. The variety of matter (biofilm, byssus, calcareous incrustations, etc.) determines a great variability in the quantity of adhered material; the mean values were found to be 14.16% and 12.92% in the Adriatic Sea and Tyrrhenian Sea samples, respectively, as shown in Table 1. 

The comparison between nets from mussel farming in the Tyrrhenian Sea and Adriatic Sea values did not display significant differences among the adhered material (paired *t*-test, t = 1.4641, *p* < 0.1402). This means that the oxidation process could be applied to each farm site using the same operating conditions.

In Figure 3, the IR spectrum of the mussel socks and the virgin PP are reported. The spectrophotometric analysis of the polypropylene highlighted the presence, on the surface, of protein materials (-NH_2_ functional groups at 3350−3180 cm^−1^ and at 1650−1515 cm^−1^) due to both the microorganisms and the mussels’ capacity to adhere themselves to various solid materials such as rocks, plastics, metals, glass and so on; this capacity depends on the presence of “glue protein” [20]. The microscopic analysis also highlighted the presence of the byssus fibres produced by mussels [21] (Appendix A). These fibres are composed of fibrous, composite-like material, with granules immersed into a protective cuticle formed by a homogeneous protein matrix [22]. These byssus fibres exhibit an amide band (1555 cm^−1^) associated with the C=O group (1655 cm^−1^). The presence of carbonates (peak at 900−1000 cm^−1^) is due to the encrustations caused by the presence of calcifying marine invertebrates (polychaete worms), sometimes very abundant (Figure 4). 

### 3.2. Effect of the Cleaner Solution

Hydrogen peroxide (H_2_O_2_) is a strong oxidant (its standard potential is 1.70 V) and has the advantage that its only decomposition product is water, so its application in the treatment of various inorganic and organic pollutants is extensively used and well documented in the literature [23]. The Fenton reagent consists of a water solution containing H_2_O_2_ and Fe^2+^ ions and is used to achieve the oxidation of persistent organic contaminants in wastewater. Generally, H_2_O_2_ is rapidly decomposed with a half-life of about 2 h if Fe^2+^ ions are present in the solution [24] and, moreover, the reaction is exothermic. H_2_O_2_ decomposition produces hydroxyl radicals (·OH), which are the agents with the highest oxidative capacity (standard potential of 2.80 V) after chlorine.

The reactions that took place in the solution are summarized below; the ferrous (II) ions were oxidized by hydrogen peroxide producing ferric (III) ions, a highly reactive ·OH, and a hydroxyl anion (OH^−^). The iron (III) was then reduced by the same H_2_O_2_ to ferrous ions, forming a radical peroxide
Fe^2+^ + H_2_O_2_ → Fe^3+^ + OH• + OH^−^(3)
Fe^3+^ + H_2_O_2_ → Fe^2+^ + OOH• + H^+^(4)

The oxidation reaction can take place both at acidic and basic pH, oxidation, however, appears more effective at acidic pH (2−4). 

The results, in terms of percentage removal of the adhered material, are shown in Figure 5; in particular, the effect of the reaction time, dimension of sock samples and the ultrasound application is exposed. The maximum removal of adherent materials was reached with the smaller sock particles (<0.5 cm) after 40 min. The application of an ultrasound strongly increases the removal, reducing the treatment time; in fact, to reach the maximum value 30 min instead 40 seems to be enough. Regarding the trend of the curves, a difference can be observed due to the dimensions of the treated fragments, in particular in the smaller samples, it is possible to observe the first rapid removal until a “plateau” is reached. 

The interaction between the adhered material and oxidant solution require that the two species come in contact; therefore, the oxidation mechanism should be considered as a bimolecular reaction. As reported in Figure 6, cleaning mussel nets using oxidation processes such as H_2_O_2_ (or HNO_3_) solutions have been shown to follow different detachment kinetic profiles according to the treatment condition. An interpretation of the kinetic profiles suggested that a one-phase model could be to fit the data relating the largest net samples (R^2^ = 0.9633 and 0.9643, respectively for 1 × 1 cm and 5 × 5 cm), in particular the diffusion of the oxidant reagent to promote the chemical detachment. A two-phase profile seems to explain the curve shape for the smallest samples, particularly if the ultrasound was applied. The change in slope may depend on multiple mechanisms, where the diffusion of the reagent, the modalities of agitation and the size of the fragments, as well as the ultrasound application and uneven distribution of the residue play a not exactly but important secondary role, which should be carefully considered during the oxidation reactor design phase.

Nitric acid is also characterized by a strong oxidizing capacity with high redox potential (0.8−0.96 V depending on the pH); the presence of small quantities of nitrous acid increases its oxidizing properties. Generally, nitric acid oxidizes organic compounds producing nitrogen oxides (NO and/or NO_2_ depending on the acid concentration) and CO_2_ according to the following equation
3C + 4HNO_3_ >> 3CO_2_ + 4NO + 2 H_2_O(5)

From the data shown in Figure 5 it can be seen that nitric acid, as a strong oxidant, can also be effectively used instead of hydrogen peroxide. Looking at the treatment plant, the only drawback may be linked to the need for off gas (NO_x_) treatment by means of a scrubber to be placed at the head of the oxidation reactor.

It should also be emphasized that the IR spectrum of the polymer treated with a raised concentration of HNO_3_ highlighted a peak attributable to functional groups of the R-O-R type (see IR spectra in Figure 7). This means that nitric acid, in high concentrations, has the ability to degrade the polymer.

### 3.3. Effect of Oxidant Concentration

The concentration of the oxidant agent is a key parameter for the detachment of organic material from nets. The effect of the oxidant dosage, H_2_O_2_ and HNO_3_ on the removal of the adhered materials is shown in Table 2. As expected, with the increase in oxidant dosage the removal of organic matter increased, in particular for the smaller fragments. In addition, according to the experimental data, material detachment further increases using ultrasound treatment (50 MHz). Considering hydrogen peroxide consumption during the oxidation reaction using 30% H_2_O_2_, the analyses show that less than 60% of the reagent was consumed. Hence, the residual solution can also be recycled or the volume/solid ratio, actually V/S = 40, can be reduced. However, it should be considered that increasing the solid phase could reduce the agitation efficiency.

### 3.4. Effect of pH and Fe on the Fenton Reaction

As already mentioned, the Fenton reaction is influenced by both the pH and the presence of a catalyst (FeSO_4_). It is therefore of fundamental importance that ferrous ions are present in the solution to allow the start of the radical catalytic mechanisms, as shown in Equation (3). Its optimal dosage is a peculiarity of the Fenton process and its concentration varies according to the type of wastewater to be treated. The main task of the oxidation process in the case of mussel nets is to degrade the solid phase to facilitate its detachment from the surface. 

Tests conducted at a constant temperature (25 °C) and reaction time (30 min), with varying pH and ferrous ions concentration, confirmed the importance of optimizing these two variables. The effectiveness of the oxidation process decreases as the pH increases; in fact, the oxidation reaction carried out at pH = 2 produces, on average, better results (Figure 8).

Increasing the pH also entails the disadvantage related to the precipitation of iron hydroxide, which occurs starting from pH 3.5 to 4.0 [25]. It will be essential to take this into account when optimizing the process, since during the reaction process, the pH can change and its adjustment is necessary, as well as at the beginning. 

### 3.5. Water as Cleaning Solution

Washing with water, on both dry and wet samples (V/S = 40), has shown some effectiveness and, in particular, 38.4% of removal was observed for the smaller fragments (<5 mm). Washing with water is to be taken into consideration when defining the treatment process; the main reason should be economical, as a pre-wash with water saves on the cost of reagents. Moreover, water can be easily recycled after filtration.

### 3.6. Mechanical Removal from Dry Socks

Surface treatment of plastic is not an uncommon procedure in the industrial world; plastics can be treated to improve wettability, leading to the proper adhesion of paints, inks, coats, etc. Adhesion problems are common in materials that possess low surface energies such as polyethylene and polypropylene, considering that, normally, polyolefin has a non-polar surface, therefore low molecular attraction between the polymer and external material exists. 

The PP has low critical surface tension compared with other material such as Nylon 6-6, PET, Aluminum and glass (31, 43, 43, 500, 1000 mN/m, respectively) [26], facilitating the simple mechanical removal of adhered organic and inorganic matter.

As reported in Table 3, the removal of the adhered material is facilitated by crushing the nets; in other words, the reduction into fragments with a size < 5 mm involves a reduction in the order of 43%, especially when the material is dry. The drying and crushing phase of the nets therefore represents a fundamental point of the treatment process aimed at recycling polypropylene. In particular, the reduction into very small fragments has a double advantage: facilitate the biofilm removal by oxidation and, after the treatment, directly feed the extruder to make the new socks.

### 3.7. Characterization of the Recovered Material

The chemical integrity of the recycled PP has been extensively investigated using Fourier-transform infrared spectroscopy (FTIR). As shown in Figure 6, no significant chemical structure changes or formations of peaks associated with the oxidation of PP were observed after treatment by hydrogen peroxide (oxygen within the polymer matrix). Considering the HNO_3_ (30%) as the oxidant chemical reagent, the presence of the C-O-C group at about 1200 cm^−1^ was observed, indicating the oxi-degradation of PP.

#### 3.7.1. TGA Analysis

The thermogravimetric analysis (TGA) technique quickly provides information on the (thermal) stability of the polymers, the degradation processes and the products of the degradation.

In Figure 9, the results of the analysis are shown and the curves substantially overlap; a slight deviation can be observed in the case of a dirty plastic net, since the increase in temperature causes the decomposition of the adhered material.

If there had been evident degradation phenomena, the thermograms of recycled PP would have demonstrated a shift to a lower temperature, revealing the presence of chain scissions. The observed net flex represents the complete degradation of the polypropylene.

#### 3.7.2. Mechanical Characterization

External forces acting on the polymeric material are the cause of deformations by partly elastic and partly viscous processes; therefore, each polymeric material has characteristic values correlated to the intrinsic mechanical properties. Polypropylene has an excellent stiffness and chemical resistance.

In order to verify the integrity of the polymeric material obtained following the treatment, tensile tests were performed directly on the socks (leaving their original shape) and not on moulded specimens obtained from the fusion of the recovered polymeric materials. In this way, the execution of the test was accelerated and the strain resistance of the PP sock was directly tested. 

The results of the tensile stress–strain tests carried out are summarized in Table 4. The experimental data further confirm that the treatment does not affect the mechanical (and chemical) characteristics of the recovered polypropylene.

### 3.8. Biofilm Control

Biofilms are complex matrices composed of many different molecules such as polysaccharides, lipids, proteins and so on. Once plastics enter the marine environment, microbes quickly colonize them and, over time, the microbial community and other microscopic organisms adhere on their surface as a biofilm [27].

As expected, a relevant microbial community and diatoms (single-celled alga) were found on the surface of the sock samples. The determination of the DNA genome sequences has shown that at least 33 genes were found to be associated with the biofilm adhered to the socks. In particular, the genus *Sulfitobacter*, mostly found in marine and hyper saline environments, was the most abundant (32.8%); the second most abundant (6.1%) genus was *Psychrobacter*, which live in cold marine habitats (Appendix A).

According to the performed tests, none of the treated samples contained any detectable microorganism, therefore, oxidation by hydrogen peroxide is excellent for the degradation of the biofilm created by microbial communities.

### 3.9. The Process Scheme

Figure 10 shows the process diagram developed for the treatment process. It involves several stages:Drying of the socks. For this phase, it is not necessary to reach high temperatures; in fact, considering the temporary storage treatment, solar energy could be sufficient to remove a large amount of water.Crushing in a blade mill to reach dimensions of less than 5 mm. The tests carried out in the laboratory made it possible to estimate the amount of residue removed during this operation and the results were encouraging. This involves an in-depth study of this aspect in order to choose the most suitable mill, especially considering possible local polymer melting and, consequently, the blocking of the apparatus.Separation of pulverized residues from fragments. This operation can also be carried out by washing the fragments with water as the sludge tends to settle.Oxidation with H_2_O_2_ (or HNO_3_). Oxidation can also be performed in milder conditions, since most of the adhered materials have been removed during the subsequent phases. The use of ultrasound could bring both process and economic advantages. This is the most important step but only very simple equipment is required for chemical oxidation, including a storage vessel for chemicals (and for wastes), metering equipment, and a reactor with a stirrer to provide the suitable contact of oxidant and sock fragments. Instrumentation to control pH, temperature and stirring speed are required. Moreover, an oxidation-reduction potential electrode can monitor the oxidation process. Considering the aggressiveness of the reagents used, the material of construction for reactor, storage vessels, pumps, valves and pipework needs to be selected with care (AISI 316, PVC, PP or HDPE). In order to save the oxidant reagent, the best V/S ratio must be empirically determined.Filtration/sedimentation.Washing with water. For this phase, the complete recycling of the process water is conceivable.Filtration/sedimentation.Polymer drying and recovery before to use.

#### Treatment Cost Analysis

The treatment cost was estimated considering the process steps listed above, energy and chemicals prices related to 2021 and a plant capable of treating 300 kg/day of mussel socks. The following conditions were selected: oxidant agent H_2_O_2_ (50% solution at 0.4 EUR/L), solid NaOH (0.60 EUR/kg), energy consumption: 13 kWh (0.22 EUR/kWh), dry matter (13%) 39 kg/batch and exhausted solution 600 L/batch. The estimated cost, excluding the labour costs, is 0.222 EUR/kg of mussel socks; the environmental costs have not been considered. Once the cost of the treatment has been obtained, the cost–benefit analysis can be carried out:Landfill disposal: 0.20–0.25 EUR/kg;Virgin socks cost: 2 EUR/kg;Virgin PP cost (2021 value): 1252 EUR /ton (d = 0.905 g/cm^3^) (statista.com/statistics);Recycled PP cost: 616–862 EUR/ton;Estimated treatment cost: 0.222 EUR/kg;Income from the sale of recovered PP: 0.739 EUR/kg (average value).

Considering that the cost of the treatment process is similar to that of landfill disposal, the cost-benefit analysis suggests that the oxidation process to treat (and recover) the mussel socks can be considered an effective opportunity to save about 36.9% of the cost of the new socks. 

## 4. Conclusions

In comparison to the main aquaculture activity, mussel farming can be considered attractive and environmentally friendly due to the absence of environmental effects arising from feed and medicines used in the process. This study showed that it is possible to solve the problem of the diffusion of mussel socks by recycling the polymer material, predominantly polypropylene. The PP recycling process involves several steps such as collection, cleaning and melting to produce granules ready to make new nets (or other items) from recycling PP. The sample dimension fragmentation plays an important role, since we prove that mechanical action is much more effective with a lower sample dimension. 

The ultrasound bath technique has been tested as a biofilm removal method, but many studies have demonstrated that it should be used alongside other methods. The recycled PP can be blended with virgin PP to produce new socks or other items. The cost–benefit analysis showed that by using the developed treatment process, economic and environmental benefits can be obtained. Recycling PP to make new nets remains the best available option to treat this kind of ocean/sea waste in an eco-friendly and cost-effective way. The results of this research have provided the basis for a Life Environment Project (Life MUSCLES), recently funded by the EU (https://webgate.ec.europa.eu/life/publicWebsite/project/details/5662) (accessed on 15 July 2022). In particular, the Life MUSCLES project involves the design, construction and use of a mobile plant to treat 300 kg per day of mussel socks.

## Figures and Tables

**Figure 1 polymers-14-03469-f001:**
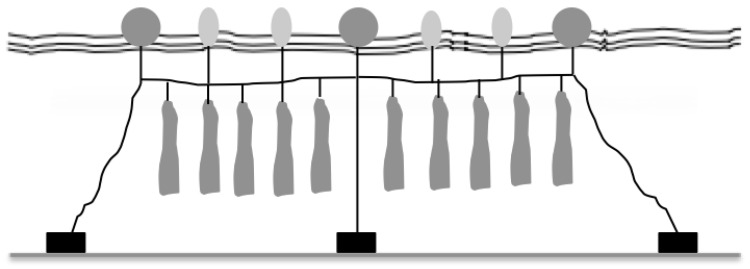
The long-line system mainly utilized in the Mediterranean Sea.

**Figure 2 polymers-14-03469-f002:**
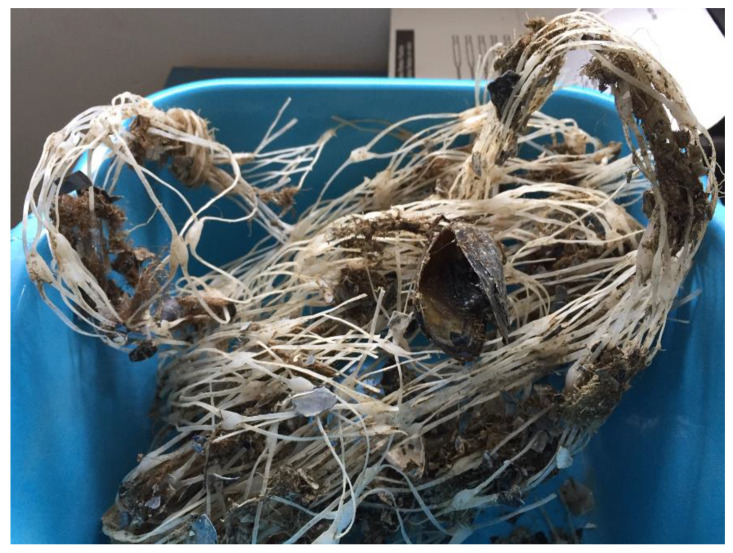
Mussel nets as recovered from the sea.

**Figure 3 polymers-14-03469-f003:**
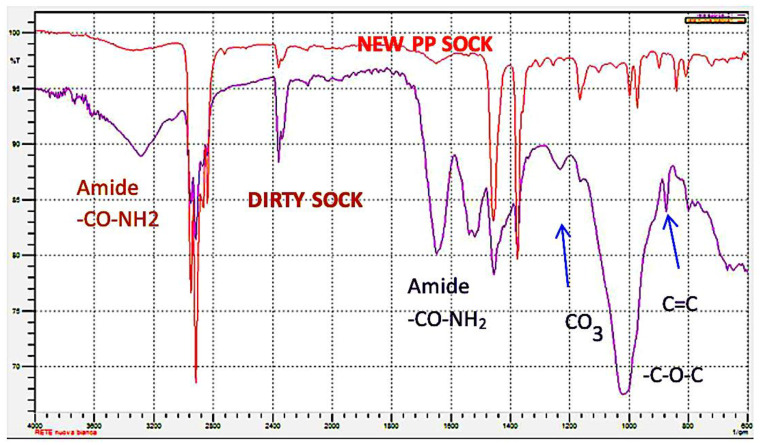
IR spectrum of new and dirty socks.

**Figure 4 polymers-14-03469-f004:**
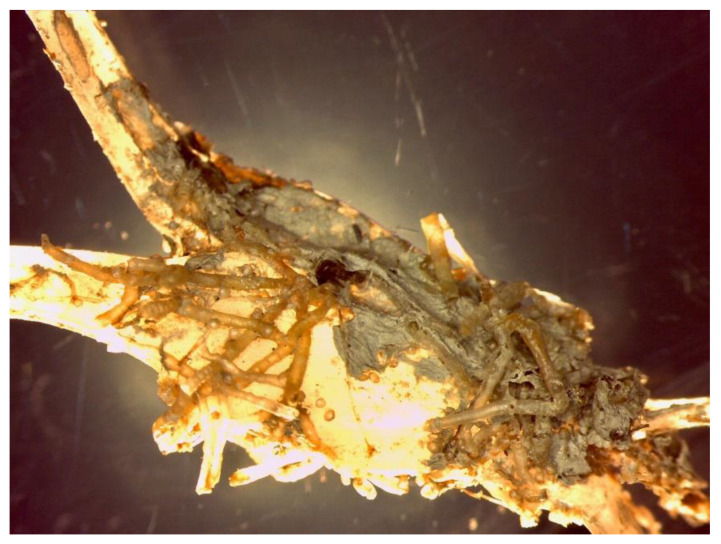
An example of the adhered materials on the sock surface.

**Figure 5 polymers-14-03469-f005:**
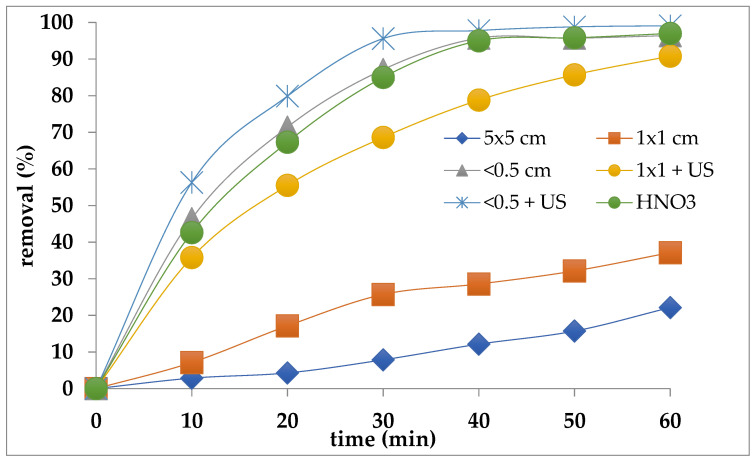
Effect of the sample dimensions on the removal of adherent materials by H_2_O_2_ = 40% and HNO_3_ (30%, samples < 5 mm). T = 20 °C, US = ultrasound 50 MHz.

**Figure 6 polymers-14-03469-f006:**
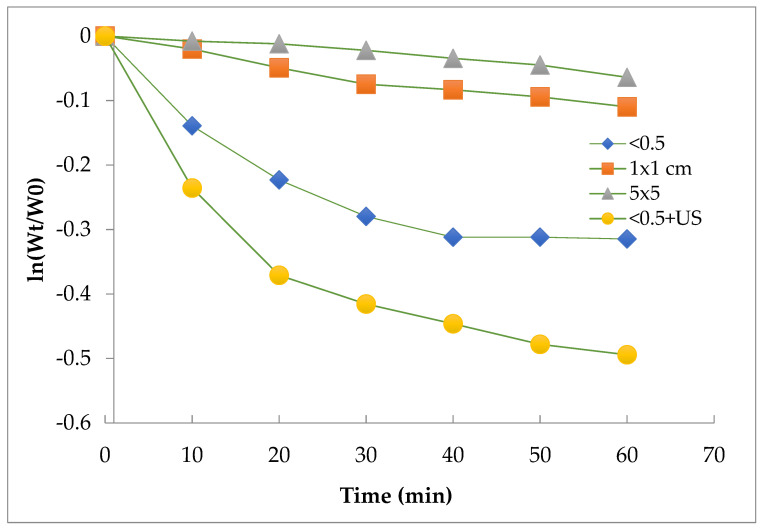
Detachment kinetics of adherent material by H_2_O_2_ (40%) under different conditions.

**Figure 7 polymers-14-03469-f007:**
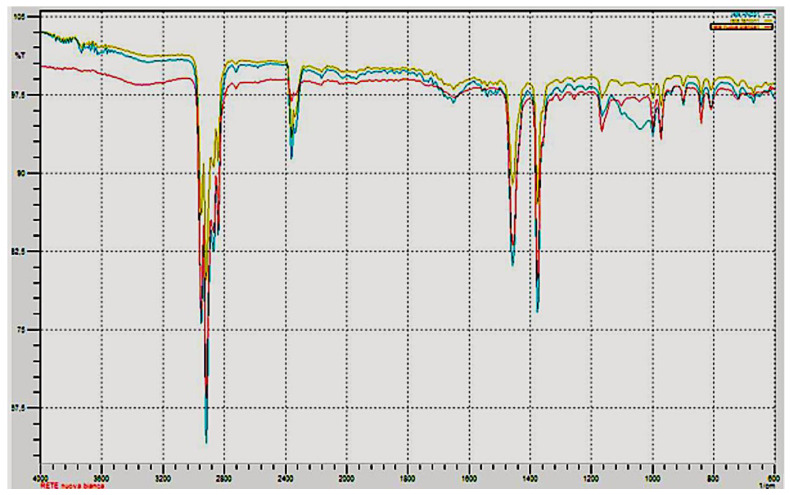
Comparison of IR spectrum of treated and new PP socks. Green = H_2_O_2_ treatment, blue = HNO_3_ treatment, red = untreated.

**Figure 8 polymers-14-03469-f008:**
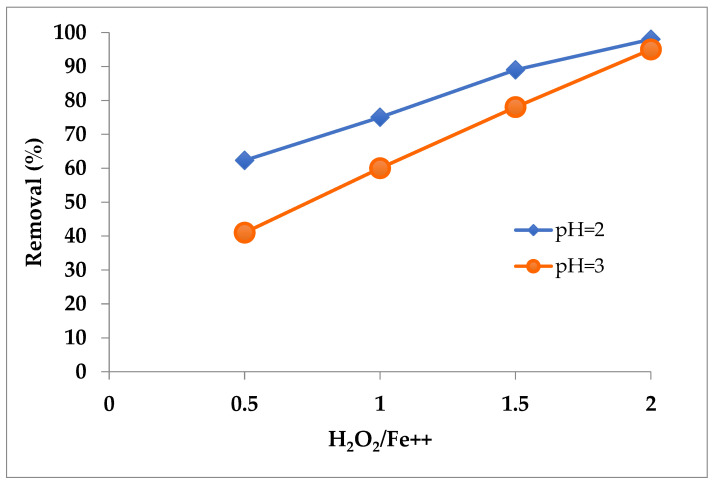
Effect of pH and Fe^2+^ concentration on the organic matter removal. T = 25 °C t = 30 min. Mean values (n = 3).

**Figure 9 polymers-14-03469-f009:**
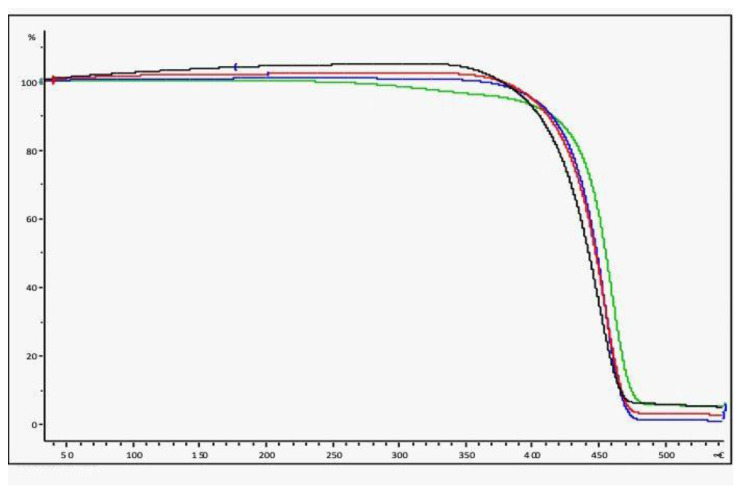
TGA analysis of the socks. Black= virgin PP, Red= treated with 30% H_2_O_2_, blue = treated with 30% HNO_3_, green = untreated socks.

**Figure 10 polymers-14-03469-f010:**
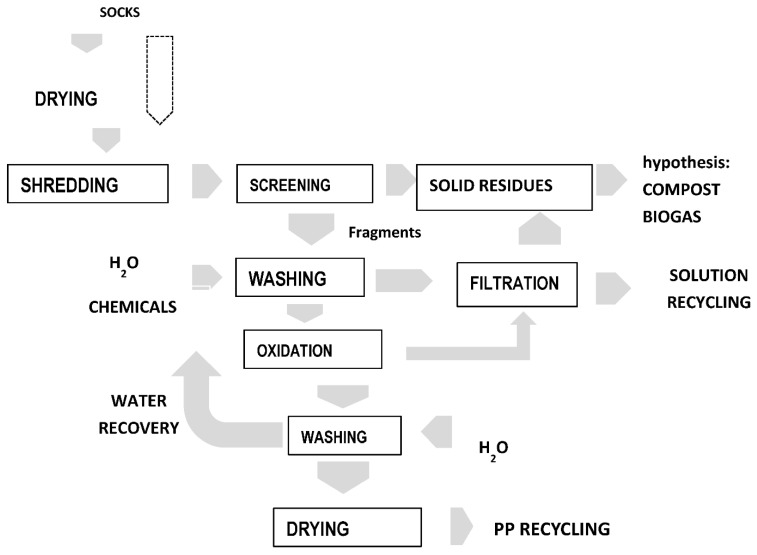
The process scheme.

**Table 1 polymers-14-03469-t001:** Adhered materials on the mussel socks (% of dry weight). N = 20 + 20.

	Adriatic Sea	Tyrrhenian Sea
Mean value	14.16	12.92
Minimum value	6.9	7.2
Maximum value	22.3	21.9
Standard deviation	3.858	3.429

**Table 2 polymers-14-03469-t002:** Effect of the oxidant concentration on the detachment efficiency (n = 3).

	H_2_O_2_			HNO_3_		
Chemical (%)	R% (<5 mm)	US R% (<5 mm)	US R% (5 × 5 cm)	R% (<5 mm)	US R% (<5 mm)	US R% (5 × 5 cm)
10	61.2 ± 1.2	78.2 ± 2.3	48.3 ± 2.1	58.2 ± 2.1	68.3 ± 1.6	41.7 ± 2.2
20	68.4 ± 2.3	86.5 ± 3.6	56.4 ± 1.8	69.3 ± 3.2	79.3 ± 2.8	52.3 ± 2.1
30	83.7 ± 2.5	95.1 ± 2.8	65.1 ± 2.5	85.6 ± 2.9	90.1 ± 3.4	59.7 ± 3.2
40	96.3 ± 3.1	98.3 ± 3.1	79.4 ± 2.6	93.4 ± 3.5	97.5 ± 1.8	73.4 ± 1.9

US = ultrasound application.

**Table 3 polymers-14-03469-t003:** Adhered material removal (%) by mechanical action during the crushing of dry nets.

Sample/Size	<5 mm	1 × 1 cm	5 × 5 cm
1	43.7	33.2	24.7
2	39.9	38.9	31.4
3	42.6	41.8	22.6
4	47.0	25.7	18.5
5	42.3	39.1	26.7
6	43.2	32.8	23.4
Mean value	43.12	35.25	24.55
+/−SD	2.31	5.87	4.32

**Table 4 polymers-14-03469-t004:** Results of the mechanical characterization of treated socks.

	Treated Sock	Virgin Sock
Elastic modules	8.43 × 10^8^ Pa	9.21 × 10^8^ Pa
Tensile strength	1.87 × 10^8^ Pa	1.82 × 10^8^ Pa
Elongation at break	0.39	0.44

## Data Availability

Data supporting the reported results can be requested from the author.

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
