# Peer review of "Polypropylene Recovery and Recycling from Mussel Nets"

_polymers, 2022, doi:10.3390/polym14173469_

Round 1
Reviewer 1 Report
This contribution examined different characteristics of the PP socks after various treatments. Overall, this contribution is interesting in terms of recycling the waste PP and turning it into a new PP for further use. Nevertheless, the organization of this contribution needs to be revised.
1. Section 3.7. In general, people would like to buy recycled PP with similar chemical and mechanical properties as those in virgin PP. Henceforth, the chemical and mechanical properties of the final recycled PP pellets/shreds as shown in Figure 9 should be examined using the standard methods, such as ASTM or ISO standards. The author didn’t explicitly report these values or use different non-standardized methods, such as the mechanical properties reported in Table 4. Further, the author should label the sample curve properly or clearly. For example, the TGA curve. Further, the DSC plot should also be shown
2. How to calculate/obtain the numbers shown in Table 3? The procedure should be shown in the Materials and Methods section in detail.
3. Table 2, The data should be presented in the format of mean plus/minus standard deviation.
4. The SEM or optical microscopic pictures for the "socks" before and after different treatment steps should be shown.
Author Response
Comment nr 1
Specific standard methods (and characteristics) for mussel socks does not exist therefore, as reported in the Materials and methods section, the ASTM D882-02 method by adapting it to the socks (new and treated by oxidation process) was used. One of the objectives of the Life project is to provide reference characteristics and to develop a standard method for determining them.
Figures were properly modified
DSC plot: unfortunately, at moment, I can’t find the DSC plot therefore the sentence has been removed considering also that the TG analysis is sufficient to certify that there have been no changes in the polymer structure
- How to calculate/obtain the numbers shown in Table 3? The procedure should be shown in the Materials and Methods section in detail.
The procedure has been added to the Mat & Met (line 157-160)
- Table 2, The data should be presented in the format of mean plus/minus standard deviation.
Done
- The SEM or optical microscopic pictures for the "socks" before and after different treatment steps should be shown.
they were reported as S1 and S2 figures, anyway fig S2 has been added to the paper text
Reviewer 2 Report
The recovery and recycling of abandoned plastic “socks” represents proper management of the waste issue coming from mussel farming. This study investigated the roles of the chemical oxidation (H2O2 and HNO3) on the destruction of organic matter from the surface of the polypropylene “socks” used in the sea farms in order to recover and recycle it. The idea is interesting and the work is very meaningful. The effects of the particle size, oxidant concentration, agitation time, and ultrasound were studied. This work can be accepted, and some minor suggestions are below.
The variable should be italic. An economical analysis of the whole process scheme could be given. And the destruction mechanism of the organic matter from the surface of the polypropylene “socks” could be discussed.
Author Response
The cost analysis evaluation and the cost-benefit analysis were added to the paper text. Although it is a very interesting argument, the evaluation of the destruction mechanism of organic (and inorganic) residues should be another scientific work. The goal of my research work was to develop a new process to recover and to recycle polypropylene in an economic (and environmental) way.
Reviewer 3 Report
The information described in the work is interesting. The following changes are required:
Line 8: change CO2 by CO2
Line 42: ….eq. per 1 kg of…? Like in line 8
Line 43: change CO2 by CO2
Line 43: change [5-7] by [5–7]
Line 52: change [5-7] by [5–7]
Line 64: delete space…6-10
Line 69: 50-60 m
Line 70: figures should appear immediately after they are mentioned in the text
Line 73: 3-7 m
Line 69-77: references missed?
Line 78: polypropylene
Line 80: 318,905 ?
Line 80: 6,378 ?
Line 84: 70,000 ? tons or tonnes?
Line 84: 1,500 ? tons or tonnes?
Line 84-86: reference missed?
Line 125: insert its chemical formula in parentheses
Line 134: indicate model, brand and country of the oven
Line 135: 60 °C
Line 140: equation 1?
Line 154: 200 mL
Line 155: 30 min at
Line 159: check the font on the line, apparently it is different
Line 162: equation 2?
Line 173,74: use the plain hyphen to indicate ranges, as in line 172
Line 176: 3200-3600
Line 177: Idem
Line 179: Idem
Line 182: 20 mL/min…like in line 104
Line 182: 5 °C/min
Line 187: mm/s
Line 213: 60 °C
Line 213: stove or oven?
Line 220: 14.16%
Line 228: ….of the polypropylene…
Line 241: -CO-NH2
Line 251: 2 h
Line 252: exothermic. H2O2 decomposition
Line 256: a highly reactive ·OH,…
Line 257: …, and hydroxyl anion (OH−). The…
Line 257: by the same H2O2 to…
Line 260: equation 3?
Line 261: equation 4?
Line 265-272: check the font on the line, apparently it is different
Line 268: 40 min
Line 269: 30 min
Line 274: HNO3; Removal; check the font on the figure, apparently it is different
Line 277: 20 °C
Line 277: <5 mm
Line 277: Mhz or MHz?
Line 284: 1x1 cm
Line 294: check the font on the figure, apparently it is different; bullets <5 mm ? 5x5 cm? <5 + US (insert spaces like in figure 1)
Line 325: the meaning of the abbreviations should be inserted under the table
Line 326: insert spaces…5 mm…5x5 cm
Line 328: H2O2
Line 328: HNO3
Line 339: 25 °C
Line 339: 30 min
Line 342: sometimes it uses the symbol (=) attached to the variable and the value, but in other examples, such as this line, it appears separately, it is necessary to standardize the format throughout the document
Line 345: 3.5-4.0
Line 348: H2O2
Line 348: check the font on the figure, apparently it is different
Line 350: 25 °C
Line 350: 30 min
Line 355: <5 mm
Line 376: <5 mm
Line 376: use centered text formatting on the values included in the table
Line 398: text appears inside the figure that was tried to be removed but still appears, I think it is necessary to remove it completely
Line 484: On the left side of the figure appears continuous numbering, which should appear on the right side, it is necessary to correct the format
Line 503: citations should not be inserted in the conclusions, these should be based on the results obtained from the investigation.
Line 522: According to the authors' guide, the title of the references must be written in lower case text format, except for the first letter of the first word. It is necessary to homogenize the correct format in the references
Line 523: According to the authors' guide, the year of the reference must be in bold text format. It is necessary to homogenize the correct format in the references
Line 523: according to the authors guide the volume should be written in cursive text format. It is necessary to homogenize the correct format in the references
Line 526: …. change 289-313 by 289–313. It is necessary to homogenize the correct format in the references
Line 527: G. Shellfish
Line 529: S.K. A global
Line 530: According to the authors' guide, the name of the journal must be inserted in its abbreviated form (italic text) in the reference. It is necessary to homogenize the correct format in the references
Line 531: J.M. Carbon
Line 533: F.M. A review
Line 533: Rev. Aquaculture, 2020, 12, 204–223.
Line 535: G. Sustainability
Line 535: scientific names should be written in italic text format; the first letter of the first word is written in uppercase, while the second word is written entirely in lowercase text format
Line 536: 2020, 12,
Note: with the examples already shown in the references section, it is necessary to make the corrections in the other references. See the format of references in the Microsoft word template of the authors guide.
Author Response
The required changes were done
Round 2
Reviewer 1 Report
The authors have properly addressed raised by the reviewer.
Reviewer 3 Report
The recommendations to the manuscript were followed